# The *Escherichia coli* transcriptome mostly consists of independently regulated modules

Anand V. Sastry [1], Ye Gao[2], Richard Szubin[1], Ying Hefner[1], Sibei Xu [1], Donghyuk Kim[1,5], Kumari Sonal Choudhary [1], Laurence Yang[1,6], Zachary A. King [1] & Bernhard O. Palsson [1,3,4]*

Underlying cellular responses is a transcriptional regulatory network (TRN) that modulates gene expression. A useful description of the TRN would decompose the transcriptome into targeted effects of individual transcriptional regulators. Here, we apply unsupervised machine learning to a diverse compendium of over 250 high-quality *Escherichia coli* RNA-seq datasets to identify 92 statistically independent signals that modulate the expression of specific gene sets. We show that 61 of these transcriptomic signals represent the effects of currently characterized transcriptional regulators. Condition-specific activation of signals is validated by exposure of *E. coli* to new environmental conditions. The resulting decomposition of the transcriptome provides: a mechanistic, systems-level, network-based explanation of responses to environmental and genetic perturbations; a guide to gene and regulator function discovery; and a basis for characterizing transcriptomic differences in multiple strains. Taken together, our results show that signal summation describes the composition of a model prokaryotic transcriptome.

[1] Department of Bioengineering, University of California San Diego, La Jolla, CA 92093, USA. [2] Department of Biological Sciences, University of California San Diego, La Jolla, CA 92093, USA. [3] Department of Pediatrics, University of California San Diego, La Jolla, CA 92093, USA. [4] Novo Nordisk Foundation Center for Biosustainability, 2800 Kongens Lyngby, Denmark. [5]Present address: School of Energy and Chemical Engineering, Ulsan National Institute of Science and Technology (UNIST), 44919 Ulsan, Korea. [6]Present address: Department of Chemical Engineering, Queen's University, Kingston, ON K7L 3N6, Canada. *email: palsson@ucsd.edu

The transcriptional regulatory network (TRN) senses and integrates complex environmental and intracellular information to coordinate gene expression of a cell. Reverse engineering the TRN informs how an organism responds to diverse stresses and unfamiliar environments[1–3]. A fully characterized TRN would enable the prediction and mechanistic explanation of an organism's dynamic adaptation to environmental or genetic perturbations.

Reconstruction of a genome-scale TRN requires a substantial number of experiments to interrogate the binding sites for each regulator and characterize their activities[4,5]. Unlike eukaryotic TRNs, which contain highly connected co-associations[6], prokaryotic TRNs exhibit a simpler structure; over 75% of genes in the model bacteria *Escherichia coli* are known targets of two or fewer transcription factors (TFs)[7].

The TRN structure is encoded in the genome as regulator-binding sites and is invariant to environmental dynamics. However, environmental and genetic perturbations alter the activity states of transcriptional regulators to change their DNA-binding affinity[8], which in turn modulates the transcriptome in a condition-specific manner[9]. Thus, a measured expression profile reflects a combination of the activities of all transcriptional regulators under the examined condition. This poses the fundamental deconvolution challenge of separating the condition-invariant network structure from its condition-dependent expression state on a genome scale.

Compendia of microarray expression profiles have been leveraged to infer TRNs by identifying shared patterns across gene-expression profiles, rather than using direct DNA-TF-binding information[10,11]. Many inference methods define groups of genes, or modules, with similar expression profiles that are often functionally related or co-expressed. Recently, a comprehensive review of 42 module detection methods showed that independent component analysis (ICA), a signal deconvolution algorithm, outperformed all other algorithms in identifying groups of coregulated genes[12].

ICA is a blind source separation algorithm used to deconvolute mixed signals into their individual sources and determine their relative strengths[13]. Prior application of ICA to microarray expression data[14] has identified co-expressed, functionally related gene sets[15–17] that often map to metabolic pathways[18,19]. A major advantage of decomposition-based module detection algorithms, such as singular value decomposition (SVD) and ICA, over clustering or network inference methods is that decomposition-based methods detect gene modules, while simultaneously computing the context-specific activity levels for these gene modules[12,20]. The overall expression levels, or activities, of ICA-derived gene sets have been leveraged to classify tumor samples[21–23] and connect transcriptional modules to disease states[24].

Although the aforementioned studies showed that ICA tends to identify biologically relevant transcriptional modules, the majority of gene modules remain uncharacterized, limiting the utility of ICA-derived results to interpret biological responses. Here, we overcome this limitation by applying ICA to a high-quality RNA-seq compendium for the well-characterized model organism *E. coli*. We find that 66% of ICA-derived gene sets clearly represent the effects of transcriptional regulators, and propose biological or genetic explanations for an additional 27% of gene sets (leaving only 7% uncharacterized). Our approach relies on: (1) the availability of high-quality, self-consistent, and condition-rich RNA-seq expression profiling datasets; (2) the use of ICA to concurrently identify regulator targets and activities; and (3) experimental validation through the association of inferred regulator targets with observed molecular interactions and through successful prediction of gene module activation. The elucidated TRN structure deconvolutes transcriptomic responses of *E. coli* into a summation of condition-specific effects of individual transcriptional regulators.

## Results

**ICA extracts regulatory signals from expression data.** In order to extract regulatory interactions from expression data, diverse conditions must be profiled to discriminate between the effects of transcriptional regulators. Previous studies have compiled transcriptomics data from independent research groups to study the transcriptional states and regulation of *E. coli*[10,25–28]. Even after resolving the significant normalization challenge with such disparate datasets, many sources of variation remain that obscure biological signals[29–31]. These datasets mostly contain microarray data; RNA-sequencing (RNA-seq) data yields higher quality data with less noise and larger dynamic range[32].

We therefore compiled PRECISE, a Precision RNA-seq Expression Compendium for Independent Signal Exploration. This high-fidelity expression profile compendium (median $R^2 = 0.98$ between biological replicates, see Supplementary Fig. 1a, b) comprises 278 RNA-seq expression profiles across 154 unique experimental conditions for *E. coli* K-12 MG1655 and BW25113. To assemble PRECISE, we collected and processed RNA-seq data from over 15 studies published by our research group (see the "Methods" section), comprising ~20% of all publicly available RNA-seq data in NCBI GEO[33] for *E. coli* K-12 MG1655 and BW25113 (Supplementary Fig. 1c). The datasets in PRECISE were generated in a single laboratory and obtained using a standardized protocol, with detailed reporting of experimental conditions and metadata to assist in usage as a comprehensive resource (Supplementary Data 1). This homogeneity mitigated batch effects (Supplementary Fig. 1d, e), simplifying the data-processing pipeline (see the "Methods" section).

We applied ICA to identify independent sources of variation in gene expression in PRECISE. The traditional use of ICA as a signal decomposition algorithm is illustrated in Fig. 1a. When applied to transcriptomics data, ICA decomposes a collection of expression profiles (**X**) into (1) a set of components, which represent underlying biological signals (**S**), and (2) the components' condition-specific activities (**A**) (Fig. 1b, c). Each component, represented by a column of **S**, contains a coefficient for each gene that represents the effect of a particular underlying signal on the gene's expression level. Components do not contain information on the condition-specific transcriptomic state. Conversely, ICA computes activity levels for each component across every condition in the compendium, represented by a row of **A**, to account for condition-dependent expression changes. Each expression profile is represented by the summation over all components, each scaled by its condition-specific activity (Fig. 1d, e).

ICA of PRECISE produced 92 robust components that explained 68% of the expression variation (Supplementary Fig. 1f, g). Most gene coefficients in a component were near zero, indicating that each underlying signal affects a small number of significant genes (Fig. 1f). We removed genes with coefficients below a threshold (see the "Methods" section), resulting in a set of significant genes for each component. We defined these condition-invariant sets of genes as i-modulons, since these genes were independently modulated at constant ratios across every condition in the compendium. Henceforth, we will focus our discussion on the i-modulons extracted from the independent components, and their respective activity levels as computed from ICA. We note that a gene may appear in multiple i-modulons if its expression is dependent on multiple underlying biological signals (Supplementary Fig. 1h).

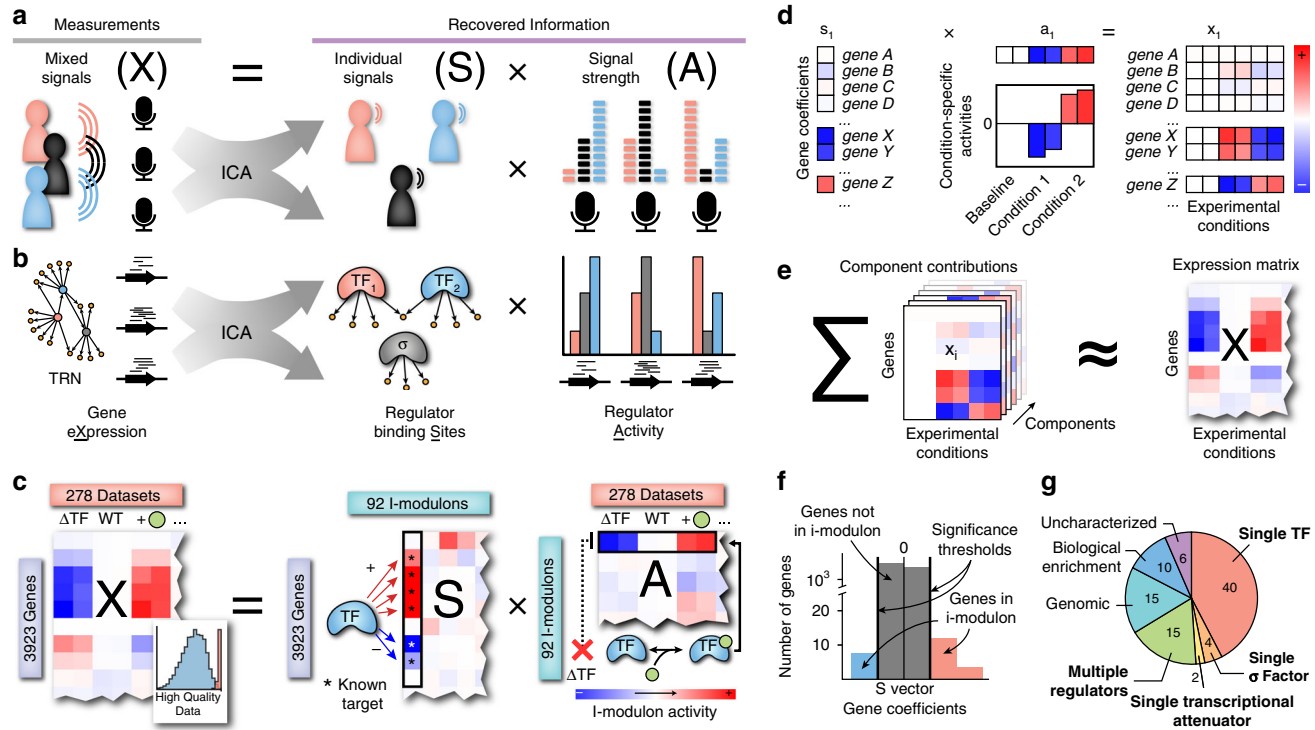

**Fig. 1** ICA extracts regulatory signals from expression data. **a** Given three microphones recording three people speaking simultaneously, each microphone records each voice (i.e. signal) at different volumes (i.e. signal strengths) based on their relative distances. Using only these measured mixed signals, ICA recovers the original signals and their relative signal strengths by maximizing the statistical independence of the recovered signals[13,71]. The mixed signals (**X**) are a linear combination of the matrix of recovered source signals (**S**) and the mixing matrix (**A**) that represents the relative strength of each source signal in the mixed output signals. This relationship is mathematically described as **X** = **SA**. **b** An expression profile under a specific condition can be likened to a microphone in a cell, measuring the combined effects of all transcriptional regulators. **c** Schematic illustration of ICA applied to a gene expression compendium. See Supplementary Fig. 1a, b for additional details on data quality. The example TF is a dual regulator that primarily upregulates genes, and is activated by the green circular metabolite. Example experimental conditions shown are a TF knock-out, wild-type, and wild-type grown on medium supplemented with the activating metabolite. Each column of **X** contains an individual expression profile across 3923 genes in *E. coli*. **d** Each component (column of **S**) contains a coefficient for each gene. These coefficients are scaled by the component's condition-specific activities (row in **A**) to form the component's contribution to the transcriptomic compendium **e**. The sum of the contributions from the 92 components reconstructs most of the variance in the original compendium. **f** Independent components are converted into i-modulons by removing all genes with coefficients within a significance threshold (indicated in gray). Significant genes may have either positive (red) or negative (blue) coefficients. **g** Distribution of i-modulon categories. Categories of regulatory i-modulons are labeled in bold font. Genomic i-modulons account for single gene knock-outs, large deletions or duplications of genomic regions. Biological i-modulons contain genes enriched for a specific function, but are not linked to a specific transcriptional regulator. For more information, see Supplementary Fig. 1 and Supplementary Table 1.

The 92 resulting i-modulons are listed in Supplementary Table 1. We hypothesized that each i-modulon was controlled by a particular transcriptional regulator, and that the i-modulon activity represented the condition-dependent activation state of the corresponding transcriptional regulator. To test this hypothesis, we examined the consistency between i-modulons and reported regulons, defined as the set of genes targeted by a common regulator, using a database of over 7000 experimentally derived regulatory interactions for *E. coli*[4] (Supplementary Fig. 1i, j).

We identified significant overlaps between regulons and 61 of the 92 i-modulons. We defined these 61 i-modulons as regulatory i-modulons (see Table 1 for a description of 30 selected i-modulons). Four regulatory i-modulons were linked to a single sigma factor (RpoS, RpoH, FliA, and FecI), two i-modulons were linked to a single transcriptional attenuator (including the thiamine riboswitch)[34–36], and 40 i-modulons were linked to a single TF. Fifteen regulatory i-modulons were associated with multiple transcriptional regulators, as described in Supplementary Table 1. Of the 31 non-regulatory i-modulons, 15 i-modulons were associated with distinct genetic changes, such as gene knock-outs and

strain-specific differences, and 10 i-modulons were enriched in a specific biological function or process (Fig. 1g).

I-modulons were manually curated, and named after their associated regulator (e.g., the MetJ i-modulon) or biological function (e.g., the Tryptophan i-modulon). The majority of regulatory i-modulons (43 of 61) mapped to metabolic pathways (Supplementary Fig. 2a), as previously noted[18]. The remaining regulatory i-modulons represented diverse cellular responses (Supplementary Fig. 2b, c). Detailed information for all 92 i-modulons, including gene composition, regulon enrichments, activity levels, and upstream regulator-binding motifs, is available in Supplementary Data 2. Additional descriptions for each i-modulon are listed in Supplementary Note 1, and code for i-modulon computation and exploratory data analysis are available on Github at https://github.com/SBRG/precise-db.

**Validation of I-modulon–Regulator relationships**. On average, 78% of genes in a regulatory i-modulon were reported targets of the linked transcriptional regulator(s) (Fig. 2a, b). In order to benchmark the precision of the i-modulons generated from

**Table 1 Summary of 30 selected regulatory i-modulons.**

| I-modulon name | Number of genes | Regulator(s) | Enrichment P-value | Precision | Recall | Activity-TF $R^2_{adj}$[a] | Biological function of I-modulon genes |
|---|---|---|---|---|---|---|---|
| ArcA-1 | 50 | ArcA | 9E−20 | 0.66 | 0.07 | 0.03 | Electron transport chain |
| Cbl + CysB | 10 | Cbl and CysB | 3E−22 | 0.80 | 0.89 | **0.73** 0.10 | Aliphatic sulfonate utilization |
| CdaR | 10 | CdaR | 2E−26 | 0.9 | 1. | **0.74** | Glucarate catabolism |
| CecR | 5 | CecR | 0 | 1. | 1. | 0.08 | Related to antibiotic sensitivity |
| EvgA | 20 | EvgA | 6E−21 | 0.5 | 0.63 | 0.06 | Acid and osmotic stress response |
| CysB | 21 | CysB | 4E−32 | 0.76 | 0.52 | 0.28 | Inorganic sulfate assimilation |
| FlhDC | 41 | FlhDC | 5E−66 | 0.93 | 0.49 | **0.72** | Flagella assembly |
| FliA | 30 | FliA | 4E−54 | 0.97 | 0.45 | **0.87** | Chemotaxis |
| Fnr | 40 | Fnr | 3E−27 | 0.85 | 0.08 | **0.57** | Anaerobic response |
| Fur-1 | 48 | Fur | 5E−55 | 0.9 | 0.25 | 0 | Iron homeostasis |
| Fur-2 | 27 | Fur | 3E−26 | 0.81 | 0.13 | 0.03 | Iron homeostasis |
| GadEWX | 17 | GadE and GadW and GadX | 1E−24 | 0.59 | 0.91 | **0.92** **0.61** **0.66** | Acid stress response |
| GlpR | 9 | GlpR | 0 | 1. | 1. | 0.02 | Glycerol catabolism |
| His-tRNA | 9 | His-tRNA attenuation | 6E−24 | 0.89 | 1. | | Histidine biosynthesis |
| Lrp | 37 | Lrp | 1E−37 | 0.86 | 0.16 | 0.13 | Amino acid and peptide transport |
| MalT | 9 | MalT | 3E−22 | 0.89 | 0.8 | **0.69** | Maltose catabolism |
| MetJ | 17 | MetJ | 2E−25 | 0.65 | 0.73 | **0.46** | Methionine biosynthesis |
| Nac | 37 | Nac | 7E−24 | 0.86 | 0.06 | **0.67** | Nitrogen starvation response |
| NarL | 29 | NarL | 1E−40 | 0.93 | 0.23 | 0.02 | Nitrate respiration |
| NtrC + RpoN | 56 | NtrC and RpoN | 3E−52 | 0.57 | 0.67 | **0.51** 0.05 | Nitrogen starvation response |
| PurR-1 | 16 | PurR | 2E−25 | 0.81 | 0.36 | 0.23 | Purine biosynthesis |
| PurR-2 | 10 | PurR | 3E−13 | 0.70 | 0.19 | 0.11 | Pyrimidine biosynthesis |
| PuuR | 7 | PuuR | 0 | 1. | 1. | **0.7** | Putrescine catabolism |
| RpoH | 13 | RpoH | 6E−20 | 1. | 0.1 | 0 | Heat shock response |
| RpoS | 107 | RpoS | 1E−18 | 0.37 | 0.13 | **0.43** | General stress response |
| SoxS | 41 | SoxS | 6E−35 | 0.56 | 0.43 | **0.74** | Oxidative stress response |
| Thiamine | 11 | Thiamine riboswitch | 0 | 1. | 1. | | Thiamine biosynthesis |
| XylR | 13 | XylR | 2E−15 | 0.46 | 0.86 | **0.83** | Xylose catabolism |
| YiaJ | 10 | YiaJ | 5E−29 | 1. | 0.91 | 0.07 | Putative ascorbate utilization |
| Zinc | 12 | ZntR or Zur | 3E−19 | 0.58 | 1. | 0 0.01 | Zinc homeostasis |

[a]$R^2_{adj}$ is computed between the activity of the i-modulon and the expression level of the TF, as described in the Supplementary Methods. Values are in bold if the $R^2_{adj}$ is above 0.4.

PRECISE, we applied the ICA workflow to 10 randomly generated subsets of a single-platform microarray compendium[10], maintaining a similar number of experimental conditions as PRECISE (see the "Methods" section). In addition, we applied the ICA workflow to a microarray dataset generated by a single research group[26] that contained a similar number of samples as PRECISE. Both the proportion and precision of regulatory i-modulons were significantly lower (Mann–Whitney $U$-test, $p$-value < 0.05) in the microarray datasets (Fig. 2c, d). We also applied sparse-PCA to PRECISE and the microarray dataset from a single research group, and found significantly less overlap with the known TRN (Mann–Whitney $U$-test, $p$-value < 0.05), illustrating that independence, rather than sparsity, was key to obtaining regulatory i-modulons (Supplementary Fig. 3a, b).

Many regulatory i-modulons contained genes that were not in the associated regulon. We hypothesized that these remaining genes were actually regulated by the associated regulator, but the binding sites were not experimentally determined. We tested this claim by performing ChIP-exo[37] to locate binding sites for the TFs MetJ and CysB (Supplementary Tables 2 and 3), which regulate methionine biosynthesis and sulfate assimilation, respectively. We identified MetJ-binding sites upstream of all 17 genes in the MetJ i-modulon (Fig. 2e). The CysB regulon was split into

the CysB i-modulon and the jointly regulated Cbl + CysB i-modulon (Supplementary Fig. 3c). We identified CysB-binding sites upstream of all genes except one (*iraD*) in both i-modulons (Fig. 2f). TF binding was not detected near 9 of the 11 genes that were in the reported MetJ and CysB regulons but not in their respective i-modulons, potentially indicating inconsistencies in previous regulon definitions. The results from these two ChIP-exo experiments showed that previously-unidentified regulator binding sites may explain the absence of genes in low-precision i-modulons (see Supplementary Note 1).

Application of ICA also extracted the condition-specific activities of i-modulons, providing an additional source of validation. I-modulon activities were centered such that all i-modulon activities were zero for a reference condition (see the "Methods" section). Thus, i-modulon activities under a particular condition represented the relative up-regulation or down-regulation of the i-modulon genes compared to the reference condition. It is important to note that positive i-modulon activities could either represent increased binding of transcriptional activators, or derepression.

In order to validate that media perturbations predictably altered specific i-modulon activities, we designed 10 expression profiling experiments to conditionally activate 20 regulators. We

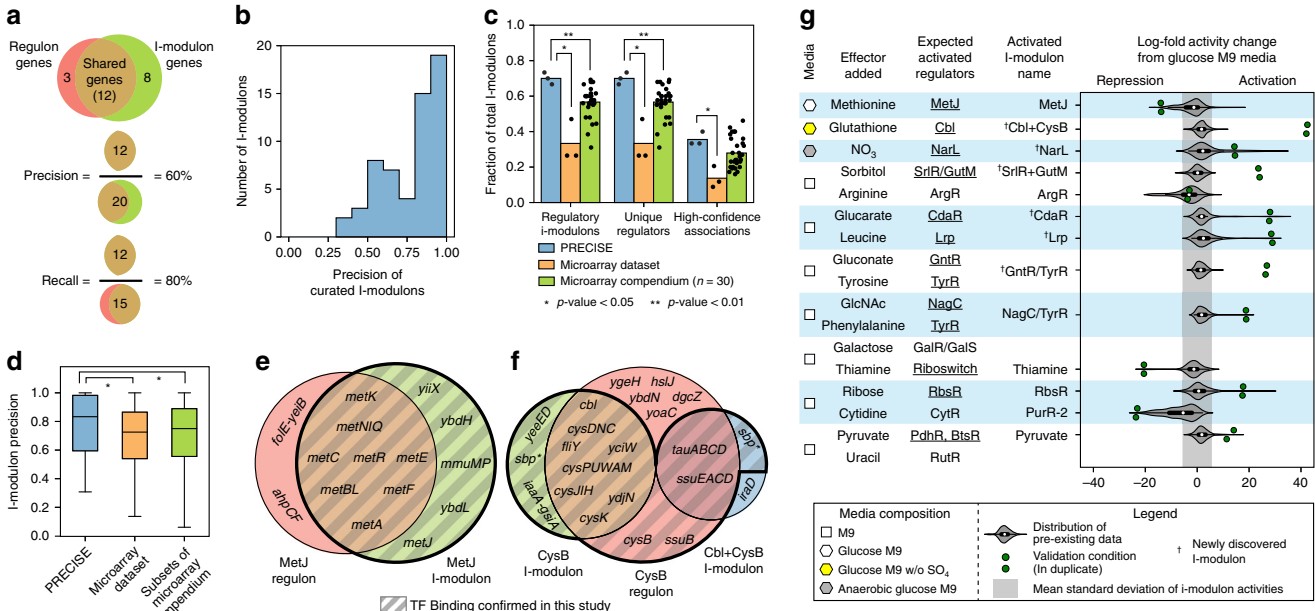

**Fig. 2** Validation of I-modulon–Regulator relationships. **a** Precision is the fraction of genes in an i-modulon that are in the linked regulon, and recall is the fraction of genes in a regulon that are in the linked i-modulon. **b** Precision across all 61 regulatory i-modulons. **c** Fraction of total i-modulons significantly enriched with targets from a single transcriptional regulator. I-modulons generated from PRECISE (blue) were compared against i-modulons generated from a microarray dataset with 266 expression profiles[26] (orange), and i-modulons generated from 10 similar-sized subsets of a single-platform microarray compendium[10] (green). Each dataset was analyzed using three-fold cross-validation (see the "Methods" section), resulting in 30 data points for the microarray compendium. Single star represents a Mann–Whitney $U$-test $p$-value < 0.05, and double star represents a $p$-value < 0.01. **d** Boxplots comparing the precision of regulatory i-modulons across all three datasets. Single star represents a Mann–Whitney $U$-test $p$-value < 0.05. Boxplot whiskers represent extrema of data, box bounds represent upper and lower quartiles, and center-line represents the median value. **e** Comparison of genes in the MetJ regulon (red) and i-modulon (green). Genes validated by ChIP-exo for co-transcribed genes were combined in the shaded regions. Gene names for co-transcribed genes were combined (e.g. *metBL* represents the transcription unit containing *metB* and *metL*) (**f**) Comparison of genes in the CysB regulon (red) and the CysB and Cbl + CysB i-modulons (green and blue, respectively). Most genes in the Cbl + CysB i-modulon were regulated by both Cbl and CysB. The starred gene, *sbp*, was a member of both i-modulons but was not in the reported CysB regulon. Genes with TF binding as determined by ChIP-exo are in the shaded regions. **g** Ten media for predicted i-modulon activations. Correctly activated i-modulons are underlined. Distribution of i-modulon activities from pre-existing data includes all data from PRECISE excluding the 10 validation conditions. The gray shaded region represents the average standard deviation across pre-existing i-modulon activities. All amino acid supplements were L-form, and all sugars were D-form. Abbreviations: GlcNAc N-acetyl-glucosamine.

confirmed 75% (15/20) of predicted activations through 13 i-modulons (Fig. 2g). The sign of the i-modulon activity revealed whether the effector resulted in a net activation or repression of i-modulon genes, which was consistent with known mechanisms for the regulators.

Seven of the 10 experiments included dual perturbations to simultaneously activate two regulators. In two cases, the two regulatory effects were recognized as a single signal, resulting in the combined i-modulons NagC/TyrR and GntR/TyrR (Supplementary Fig. 3d, e). Combined i-modulons may also occur if a single molecule activates multiple regulators (Supplementary Fig. 3f, g). Cytidine supplementation did not activate the cytidine-binding transcription factor CytR; however, the pyrimidine biosynthesis PurR-2 i-modulon was activated. Although four media additions (arginine, cytidine, galactose, and uracil) did not activate regulon-enriched i-modulons over the reference condition, the i-modulon structure of the TRN proved robust to additional data and displayed predictive capabilities.

**ICA reveals independent modules within the PurR regulon.** In order to gain a detailed understanding of the biological roles of individual i-modulons, we programmatically generated a summary of characteristics for each i-modulon (see Supplementary Data 2). Figure 3 demonstrates the biological understanding to be gained from these characteristics for two exemplary i-modulons with significant overlap with genes in the PurR regulon (Fisher's

exact test $p$-values < 1e−10), named PurR-1 and PurR-2, respectively.

PurR is a repressor of nucleotide biosynthetic genes and is activated by intracellular purine levels[38]. The PurR-1 i-modulon contained 16 significant genes with both positive and negative coefficients, of which 14 were related to purine metabolism (see Supplementary Fig. 2a). The PurR-2 i-modulon contained 10 genes, of which nine were in the pyrimidine biosynthetic pathway (Fig. 3a). Together, the two PurR-related i-modulons accounted for 20 of the 36 genes in the reported PurR regulon (Fig. 3b). Segmentation of regulons into multiple constituent i-modulons was observed for other global regulators such as Fur and Crp (Supplementary Fig. 4a, b).

The 14 genes with positive coefficients in the PurR-1 i-modulon were associated with purine biosynthesis, with 13 genes confirmed to be regulated by PurR. We detected the PurR binding motif upstream of the missing gene (*ghxP*), suggesting that it is regulated by PurR (Fig. 3c). Similar analysis identified 150 previously unidentified regulator-binding sites across 20 regulatory i-modulons (Supplementary Table 4). The two genes with negative coefficients (*add* and *ydhC*) were expressed inversely to the activation of the purine biosynthetic pathway; *add* encodes the first enzyme in the purine-degradation pathway, and *ydhC* is a putative transporter. Since the negative gene coefficient indicated that *ydhC* expression responded inversely to purine biosynthetic gene expression, we hypothesized that *ydhC* was a purine-related

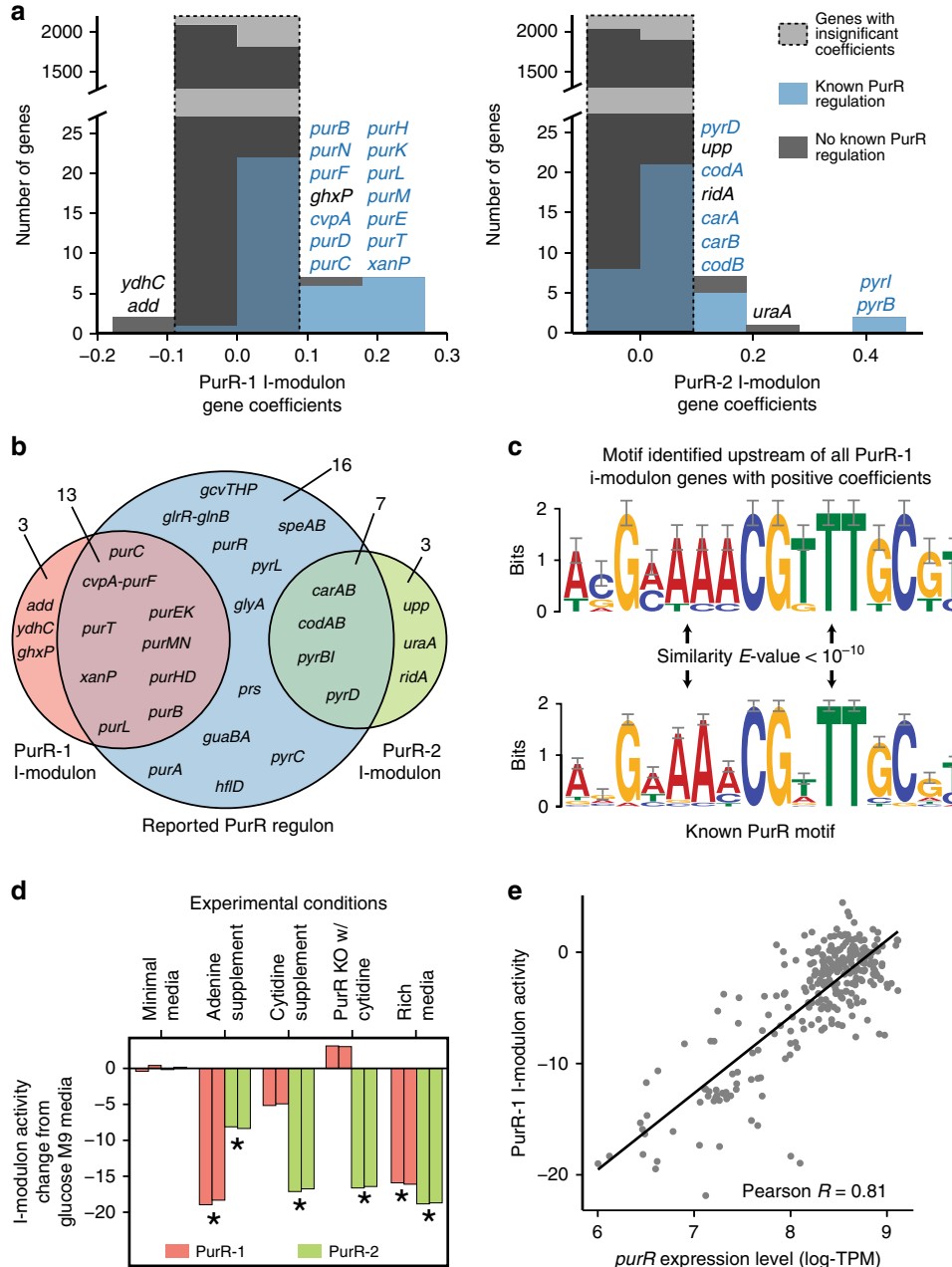

**Fig. 3** ICA reveals independent modules within the PurR regulon. **a** Histograms of gene coefficients in the PurR-1 and PurR-2 i-modulons. **b** Comparison of genes in the reported PurR regulon (blue), PurR-1 i-modulon (red) and PurR-2 i-modulon (green). Gene names for co-transcribed genes were combined (e.g. *codAB* represents *codA* and *codB*). **c** Motif identified upstream of genes in the PurR-1 i-modulon compared to the reported PurR motif from RegulonDB[4]. This motif was identified upstream of the guanine/hypoxanthine transporter encoding gene *ghxP*, although regulator binding was not previously reported. **d** The two PurR-associated i-modulons exhibited distinct responses to environmental perturbations. Asterisks denote significant i-modulon activities as compared to the reference condition (see the "Methods" section). Each bar represents a single biological replicate. **e** The PurR-1 i-modulon activity level is highly correlated with *purR* expression level across all conditions (excluding the PurR knock-out), whereas the PurR-2 i-modulon activity exhibits poor correlation (see Supplementary Fig. 4d). Similar information on all 92 i-modulons is available in Supplementary Data 2. Abbreviations: log-TPM log-transformed transcripts per million.

efflux pump. In a similar fashion, we used the i-modulon structure to generate additional information for 224 genes with poor annotations, including 11 transporters (Supplementary Table 5). One such prediction, *yjiY*, was recently independently verified as a pyruvate transporter and renamed to *btsT*[39].

The condition-specific activities of the PurR-related i-modulons revealed differences in purine and pyrimidine biosynthetic gene expression (Fig. 3d). Adenine supplementation activated the repressor PurR to decrease the activity of the PurR-1 and PurR-2

i-modulons, whereas cytidine supplementation resulted in a decrease in PurR-2 i-modulon activity. Knock-out of PurR did not affect the PurR-2 i-modulon activity, while de-repressing the purine biosynthetic pathway. LB rich medium decreased both i-modulon activities.

The relationship between the quantitative activities of the PurR i-modulons and the expression level of *purR* indicated the drivers of regulator activity. The activity of the PurR-1 i-modulon was highly correlated (Pearson $R = 0.81$, $p$-value $< 10^{-10}$) with the

expression level of *purR* (Fig. 3e). A similar relationship was observed in 26 of 61 regulatory i-modulons (Supplementary Table 1), 20 of which required a minimum expression level to activate the i-modulon (See Supplementary Fig. 4c). In contrast, the activity of the PurR-2 i-modulon was poorly correlated with *purR* expression (Pearson $R = 0.3$, *p*-value $= 6*10^{-7}$), and was likely controlled by UTP-dependent reiterative transcription[40] (Supplementary Fig. 4d). The low correlation between many i-modulon activities and their corresponding regulator expression levels specifically identified where static TRNs are inconsistent with expression data[41].

The results presented in this section demonstrate that the i-modulon structure of the TRN revealed by ICA provides a deep understanding of its biological functions and offers a guide to discovery.

**ICA provides answers to unasked questions**. Most expression profiles in PRECISE were designed to test specific biological hypotheses leading to independent, self-contained studies. However, by integrating these datasets together and applying ICA, we gained the ability to answer questions not addressed in the individual studies.

To illustrate this concept, we examined the i-modulons activated by the addition of 80 new expression profiles from four recently published RNA-seq datasets[42–45] (Fig. 4a). The expression profiles were generated and processed using the same protocol as the rest of the compendium. ICA of the original 198 expression profiles resulted in 73 i-modulons, 65 of which were maintained upon incorporation of the 80 additional expression profiles. Analysis of the i-modulons resulting from three nested subsets of PRECISE confirmed that i-modulons were rarely lost upon addition of new data (see the "Methods" section, Fig. 4b). The 80 additional datasets activated 26 new i-modulons, including 14 regulatory i-modulons and 9 genomic i-modulons. The activities of each new i-modulon revealed the causative expression profiles (Fig. 4c).

For example, one dataset included knock-outs of 10 uncharacterized TFs[42], which generated five new i-modulons. Two i-modulons were dominated by a single knocked-out gene (Supplementary Fig. 4e), whereas two remaining i-modulons (YiaJ and YieP) primarily contained genes with experimentally determined TF-binding sites (Fig. 4d, e), revealing two new high-confidence regulons. Knock-out of *cecR* produced an i-modulon that was recently independently validated with high accuracy[46]. Since the genes in the three new regulatory i-modulons had positive coefficients and the i-modulon activity was positive upon gene knock-out (Fig. 4c), we concluded that the TF knock-out resulted in overexpression of these genes. Therefore, we characterized these TFs as transcriptional repressors.

The other three datasets contained endpoint strains from adaptive laboratory evolutions (ALEs)[43–45]. ALE endpoint strains often contain many mutations whose downstream effects are difficult to resolve[47]. However, most strains contained at least one mutation in or near a transcriptional regulator that corresponded with explainable changes in i-modulon activities (Fig. 4c, Supplementary Table 6). Most of these regulators were well-characterized repressors and resulted in derepression of the i-modulon genes.

However, two i-modulons were activated in strains, where mutations occurred within TFs with no published information (YgbI and YneJ). The YneJ i-modulon contained six genes, of which *proQ* and *entC* are knocked-out in the strain with highest i-modulon activities (Fig. 4f). Three remaining genes were divergently transcribed from *yneJ*, hinting at a potential regulator binding site.

The YgbI i-modulon contained nine genes, five of which were divergently transcribed from *ygbI*, and had putative functions for four-carbon sugar catabolism[48]. The YgbI i-modulon was activated during adaptation of *E. coli* MG1655 to growth on m-tartrate, a four-carbon sugar. Although *E. coli* does not support native growth on m-tartrate, the evolved strains were able to utilize m-tartrate as the primary carbon source. Of the remaining four genes, two may be explained by other mutations present in this endpoint (see Supplementary Note 2). Therefore, we hypothesize that YgbI is a transcriptional repressor that binds between *ygbI* and *ygbJ* (Fig. 4f).

ALE strains often contain deletions or duplications of large regions in the genome, which are detected as genomic i-modulons. The Deletion-1 i-modulon resulted from a 39-gene deletion (Fig. 4g), which included KdgR, a transcription factor known to repress *kdgT* and *kdgK*[49]. These genes, along with *asr*, *hyi*, *kduD*, and *kduI*, had negative i-modulon coefficients, indicating that they were overexpressed when KdgR was knocked out. Since *kduD* and *kduI* encode genes with similar metabolic functions to *kdgK* (Supplementary Fig. 4f), we predict that they are also repressed by KdgR (Fig. 4f).

When new experimental data was added to the existing compendium, answers were revealed to questions unasked by the original studies. Next, we show that correlations between i-modulon activity levels can do the same by revealing relationships between the cellular functions represented by the i-modulons.

**Two i-modulons characterize the 'Fear vs. Greed' Tradeoff**. Can the i-modulon decomposition be utilized to understand a major genetic perturbation of the transcriptome? Adaptive laboratory evolution of *E. coli*[50] revealed two distinct point mutations in RpoB, the RNA polymerase subunit *β*, that shift cellular resources towards growth-related functions (i.e. greed) away from stress-hedging functions (i.e. fear)[51].

The quantitative i-modulon activities for two RpoB-mutant strains reflected this trade-off (Fig. 5a). Six i-modulons whose activities significantly deviated from the wild-type strain were initially uncharacterized. Further investigation into one of these i-modulons revealed genes encoding translation machinery, such as ribosomal proteins, to comprise one of the uncharacterized i-modulons (Fig. 5b). The compendium-wide activity of this Translation i-modulon was correlated with growth rate (Pearson $R = 0.59$, *p*-value $< 10^{-10}$, Supplementary Fig. 4g), consistent with previous observations that growth is propelled by increased ribosomal catalytic activity[52–55]. The Translation i-modulon therefore represented the greedy, growth-related functions of the transcriptome.

The i-modulon with the largest activity decrease in both variants was enriched in genes controlled by the stress response sigma factor (RpoS). The RpoS i-modulon activity was correlated (Pearson $R = 0.65$, *p*-value $< 10^{-10}$, Supplementary Fig. 4h) with the expression level of *rpoS* and revealed a quantitative measure of cellular stress across diverse conditions (Fig. 5c). Therefore, the RpoS i-modulon represented the fearful stress-hedging functions of the transcriptome.

PRECISE also contained expression data for over 45 strains from ALE experiments[47], many of which contained mutations in genes encoding RNA polymerase subunits. All strains with mutated *rpoB* or *rpoC* genes exhibited low RpoS i-modulon activities, reflecting a reduction in stress-related expression. Further examination revealed that the RpoS i-modulon activity was anti-correlated with the Translation i-modulon activity (Pearson $R = -0.60$, $p < 10^{-10}$), illuminating the compendium-wide transcriptomic trade-off between fear and greed (Fig. 5d). The mutations in *rpoB* shifted the strains

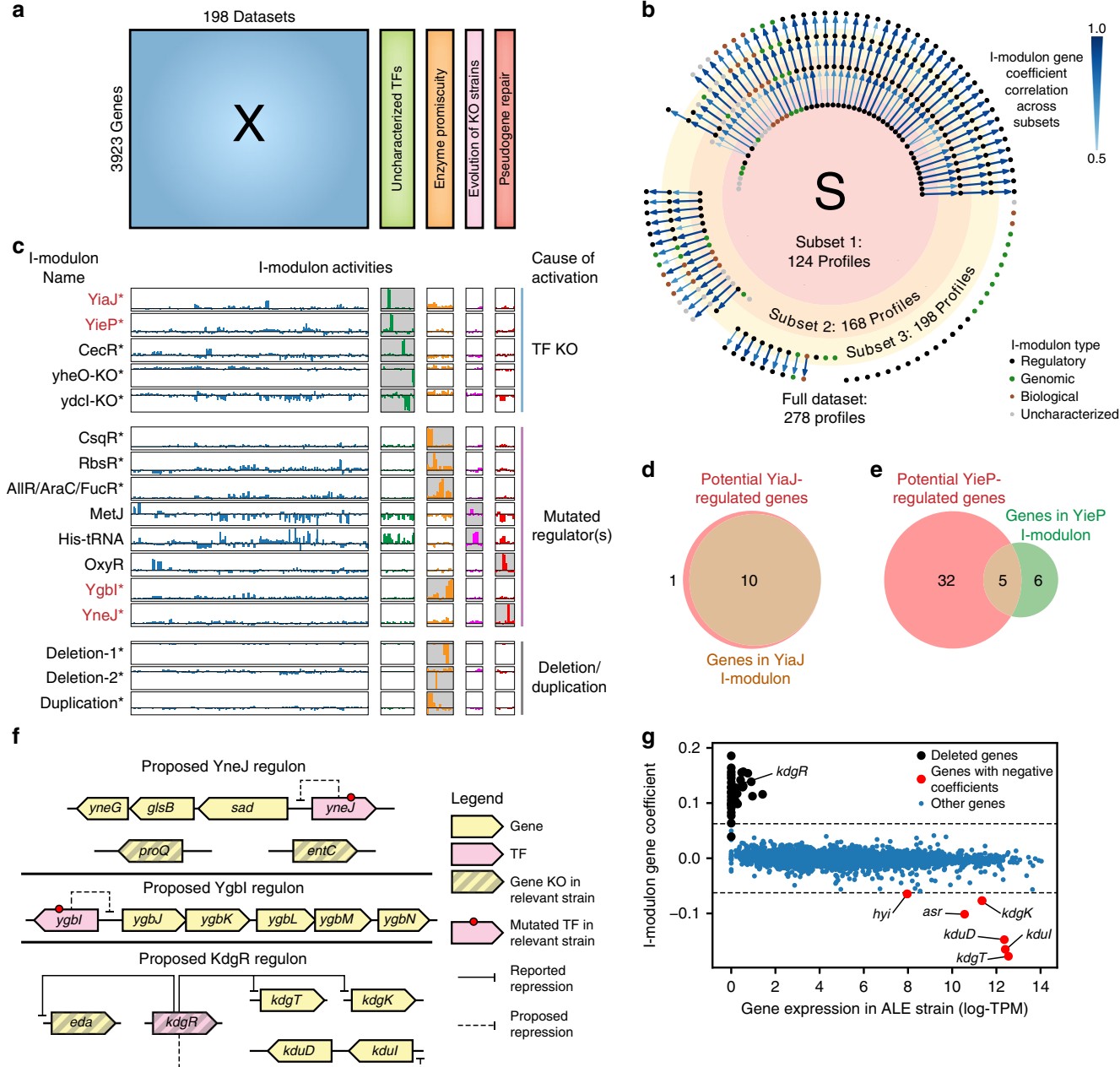

**Fig. 4** ICA provides answers to unasked questions. **a** Schematic illustration of appending four new datasets[42–45] to PRECISE. **b** Comparison of ICA results on three nested subsets of the PRECISE compendium. Each node represents an i-modulon and is colored by type (e.g. regulatory or genomic). Components are linked by an arrow if their gene coefficients are correlated (Pearson $R > 0.5$). Arrow widths and color represent correlation strength. **c** Compendium-wide activities for selected i-modulons. Each bar represents the activity of the denoted i-modulon in a single expression profile. Starred i-modulons were discovered after addition of the four new datasets. I-modulons in red font propose regulons for previously uncharacterized TFs. I-modulons are grouped based on the genetic perturbation (e.g. TF KO, mutation in regulator) that activated the specific i-modulon. The dataset responsible for the i-modulon activation is highlighted in gray for each i-modulon. **d** Venn diagram comparing genes in the YiaJ i-modulon and genes with ChIP-exo determined binding sites for YiaJ. **e** Venn diagram comparing genes in the YieP i-modulon and genes with ChIP-exo determined binding sites for YieP. **f** Predicted regulatory roles based on i-modulons for YneJ, YgbI, and KdgR. **g** Scatterplot of gene expression in strain with 39-gene deletion against the Deletion-1 i-modulon gene coefficients. The Deletion-1 i-modulon has a negative activity for the strain with the deletion, indicating that genes with positive i-modulon coefficients are not expressed in this strain, whereas genes with negative i-modulon coefficients are over-expressed in this strain.

along this line, increasing growth and reducing stress-related gene expression.

The results presented in this section show that the fear-greed tradeoff, and other transcriptome-restructuring events, can be studied in great detail by decomposing the transcriptome into a summation of independent regulatory events.

**An i-modulon discriminates between two _E. coli_ strains.** The PRECISE compendium includes 46 RNA-seq datasets from _E. coli_ BW25113, a closely related strain to _E. coli_ MG1655. The BW25113 strain was the background strain for the Keio collection of over 3000 single-gene knock-outs[56]. The transcriptomic differences between these strains, resulting from 29 genetic

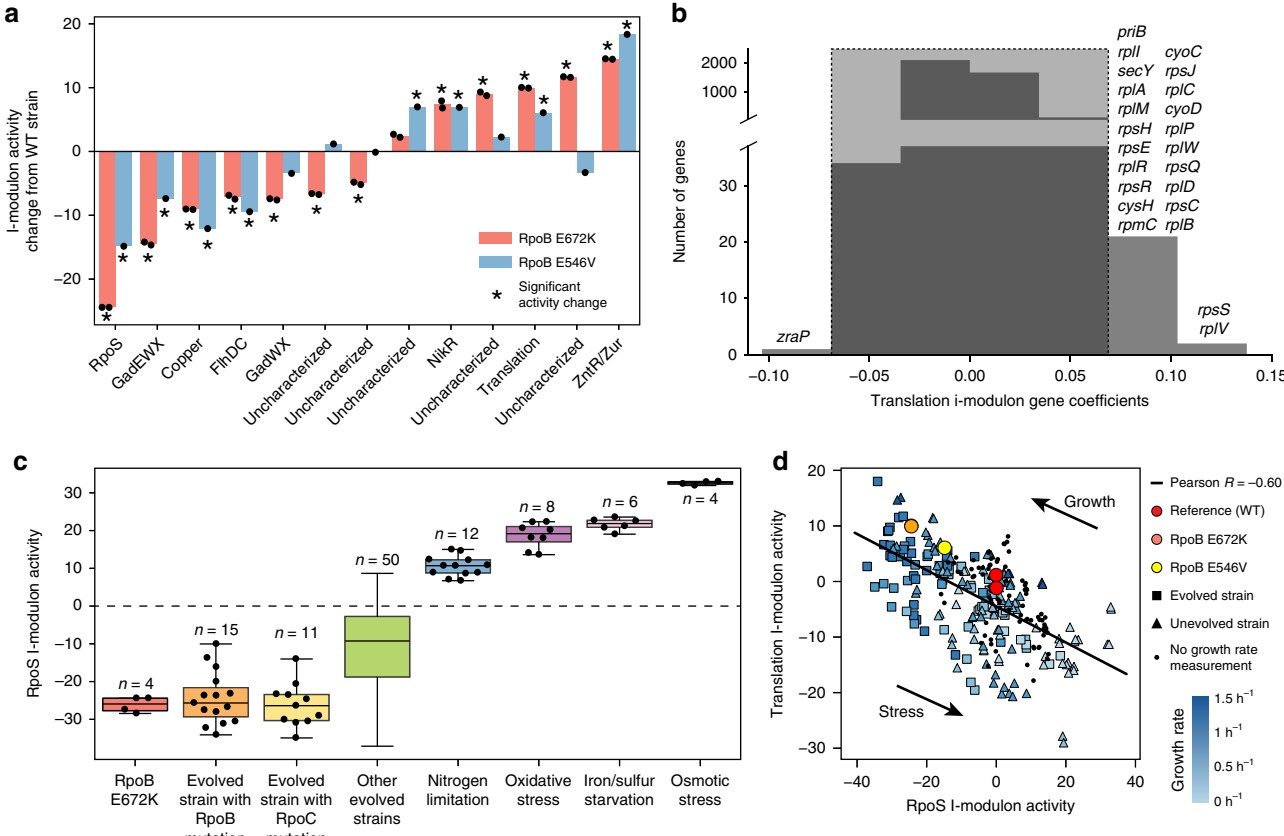

**Fig. 5** Two i-modulons characterize the 'Fear vs. Greed' Tradeoff. **a** Comparison of i-modulon activities in the RpoB E672K and RpoB E546V mutant strains grown on glucose minimal media against wild-type activities. Significant i-modulon activities are designated by asterisks (see the "Methods" section). For detailed information about these i-modulons, see Supplementary Data 2. **b** Histogram of translation i-modulon gene coefficients. Gene names are shown for genes above threshold. **c** The RpoS i-modulon activities revealed the stress level of the cell under various conditions. Boxplot whiskers represent extrema, box bounds represent upper and lower quartiles, and center-line represents the median value. **d** The RpoS i-modulon activities were anti-correlated with the Translation i-modulon activities, highlighting the trade-off between stress-hedging and growth. Single nucleotide mutations in RpoB (in yellow and orange) shifted cellular resources along this line from the wild-type strain (in red). Points were colored by growth rate measurements when available.

variations[57], have not been characterized. We identified a single i-modulon whose activities separated the transcriptomes of the two strains (Fig. 6a). The i-modulon gene coefficients elucidated nine transcriptomic differences explained by genetic differences between the strains (Fig. 6b and Supplementary Table 7).

We then sought to use the summation of i-modulons to quantitatively account for these strain-specific differences. To this end, we analyzed two expression profiles in the compendium: the reference condition (wild-type *E. coli* MG1655) and *E. coli* BW25113 with thiamine and ferric chloride supplementation. Only two i-modulons were differentially activated between the two conditions: the BW25113 i-modulon (described above) and the Thiamine i-modulon that controls thiamine biosynthesis through a riboswitch[34]. We accounted for the transcriptomic differences by subtracting the gene coefficients in the two i-modulons scaled by their respective i-modulon activities from the *E. coli* BW25113 expression profile. This subtraction increased the $R^2$ between the two strains' expression profiles from 0.29 to 0.96 for the BW25113 i-modulon genes, and from below 0 to 0.95 for the Thiamine i-modulon genes. (Fig. 6c). A similar increase in the $R^2$ value for the BW25113 i-modulon genes was observed, when the i-modulon was subtracted from any BW25113 expression profile. This illustrated that the summation of the independently modulated genes captured the major expression differences between the two strains.

In the previous section, we used the i-modulon structure to interpret the effects of a single mutation in *rpoB* on the transcriptome. Here, we illustrated the broader ability of the i-modulons to interpret and quantitatively account for the transcriptional differences between closely related strains. Next, we evaluate if i-modulons can provide understanding of larger scale genetic differences between strains.

**I-modulons explain expression changes across *E. coli* strains**. It has proven difficult to compare transcriptional regulation between different strains of the same species[58]. We examined the ability of i-modulons to provide a structured basis for such comparison. We grew and expression profiled a set of eight diverse *E. coli* strains in identical media (with additional supplements for BW251123 as described above) and calculated their i-modulon activities using the 92 previously identified i-modulons as a basis (Fig. 6d). Three strains (MG1655, BW25113 and W3110) diverged from the same ancestral K-12 strain with limited genetic differences; CFT0173 and O157: H7 EDL933 are pathogenic strains; and the remaining three strains (BL21(DE3), HS, and Crooks) are laboratory strains.

The expression profiles of the K-12 strains shared similar i-modulon activities, including higher activities in the pyrimidine-

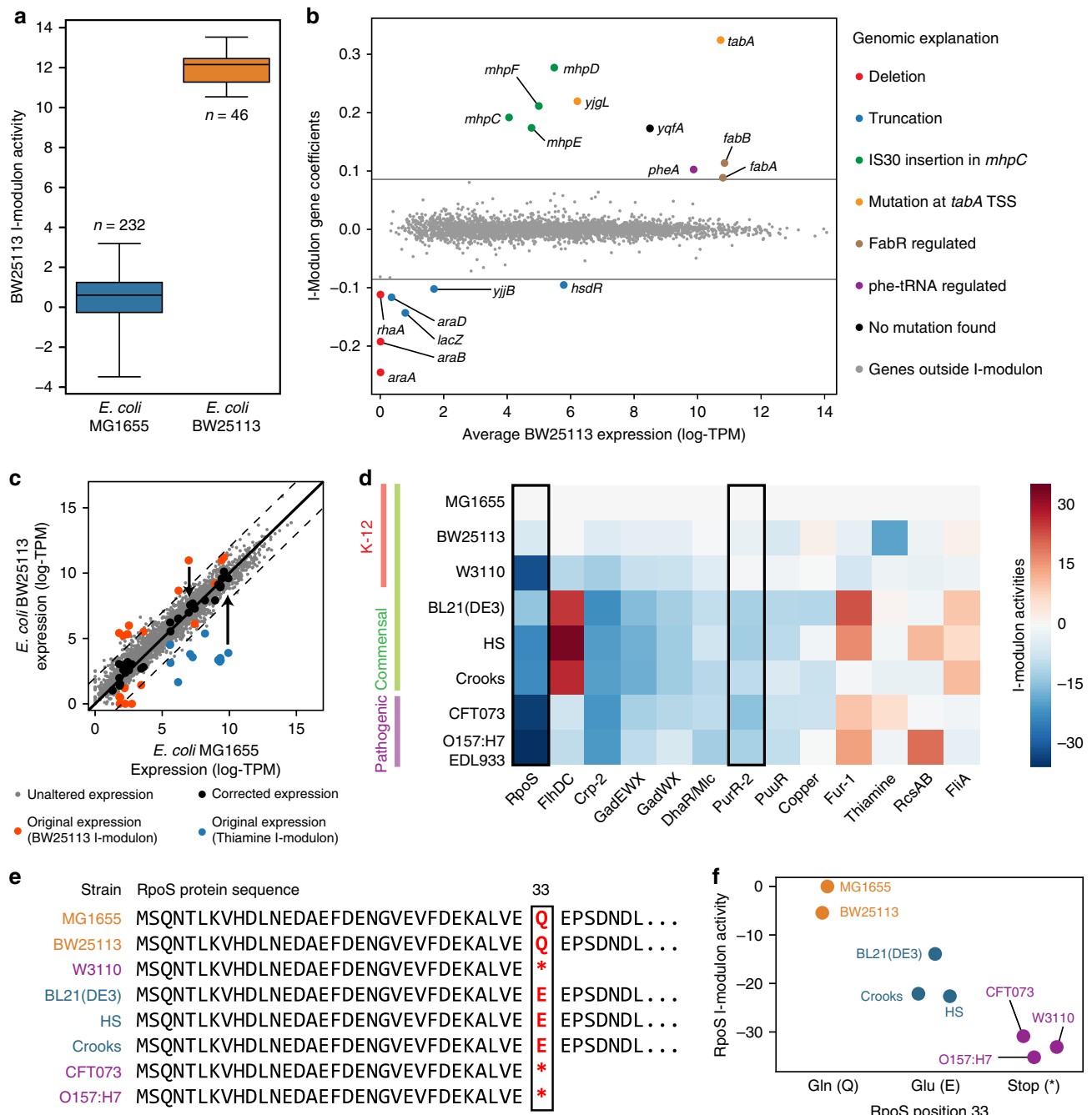

**Fig. 6** I-modulons identify differences in transcriptional regulation across multiple *E. coli* strains. **a** Boxplot of BW25113 i-modulon activities separated by strain. Number of expression profiles in PRECISE from each strain is shown. Boxplot whiskers represent extrema of data, box bounds represent upper and lower quartiles, and center-line represents the median value. **b** Scatterplot of average BW25113 expression against BW25113 i-modulon activity. Deletions and truncations in the BW25113 strain account for all genes with negative coefficients. An insertion sequence (IS30) in the *mhpC* gene in the BW25113 strain corresponds to a large increase in expression of *mhpCDEF*, as IS30 contains a known promoter[79]. Point mutations at the predicted transcription start site (TSS) of *tabA*, in the FabR regulator, and in the phenylalanine tRNA *pheV*, account for other genes with positive coefficients (see Supplementary Table 7). **c** Subtraction of the BW25113 and Thiamine i-modulons from the *E. coli* BW25113 expression profile accounts for the major transcriptomic deviations from *E. coli* MG1655 grown without thiamine. Dashed lines indicate four-fold difference in TPM. **d** Heatmap of estimated i-modulon activities for eight *E. coli* strains grown on glucose minimal media (with added thiamine and ferric chloride for BW25113). Only significantly altered regulatory i-modulon activities are shown. Boxed i-modulon activities are referred to in the main text. **e** Sequence alignment of the RpoS protein across the eight *E. coli* strains. **f** RpoS activities of the eight strains grouped by position 33 in the RpoS protein sequence, as detailed in panel **e**. Abbreviations: TSS transcription start site.

responsive PurR-2 i-modulon. This increased expression can be explained by a defect in the *rph-pyrE* operon, which leads to pyrimidine starvation in the K-12 strains[59]. The RpoS i-modulon activity was significantly suppressed in all strains except MG1655

and BW25113. Strains W3110, CFT073, and O157:H7 EDL933 have an amber mutation that results in a truncation in *rpoS* and is known to reduce its expression and activity[60,61]. Three other strains (BL21 (DE3), HS, and Crooks) contain a divergent mutation at the same

position that likely explains their reduced RpoS i-modulon activity (Fig. 6e, f).

These results demonstrate that the i-modulons derived from a single strain can provide a scaffold to analyze transcriptomes from other strains of the same species. Strain-specific differences in i-modulon activities can be traced to sequence variations in the associated regulators, thus providing a deep explanation for targeted strain differences.

## Discussion

We have demonstrated that the combination of (1) ICA of high-quality RNA-seq data and (2) high-resolution comprehensive regulator-binding site information, identifies linear combinations of quantitative regulatory signals that reconstitute the *E. coli* transcriptome, leading to the first *E. coli* TRN inferred from an RNA-seq compendium. This result suggests that a principle of i-modulon addition governs the composition of the *E. coli* transcriptome. Application of this principle provided a multi-dimensional understanding of the *E. coli* TRN, and uncovered detailed responses to environmental and genetic perturbations, optimality of adaptation to new conditions, and links between genotypes and phenotypes of *E. coli* strains. If this principle governs the composition of other prokaryotic transcriptomes, it provides a path to develop a detailed understanding of transcriptional regulation in less-understood organisms.

We have shown that for the model prokaryote *E. coli*, 61 of the 92 identified i-modulons represent the effects of characterized transcriptional regulators. This coverage is a consequence of the quality and diversity of the RNA-seq compendium used, the extensive information available on *E. coli* transcriptional regulators, and the relative simplicity of the *E. coli* TRN. In principle, if we obtained expression data for every condition sensed by a prokaryote, we could decompose its expression state to a non-reducible set of regulatory signals. These fundamental signals, combined with high-throughput-binding data for all TFs in an organism[42], would lead to the establishment of a comprehensive quantitative TRN.

## Methods

**RNA extraction and library preparation.** Total RNA was sampled from duplicate cultures. All strains were grown in minimal salts (M9) medium at exponential phase, with complete growth conditions listed in Supplementary Data 1. Growth curve analysis were performed using Bioscreen C Reader system with 200 μL culture volume per well. Two biological replicates were used in the assay. Media components were purchased from Sigma-Aldrich (St. Louis, MO). For nitrate respiration cultures, a 35:50 ratio of carbon dioxide to nitrogen was bubbled through the media to deoxygenate. After inoculation and growth, 3 mL of cell broth ($OD_{600} \sim 0.5$) was immediately added to two volumes Qiagen RNA-protect Bacteria Reagent (6 mL), vortexed for 5 s, incubated at room temperature for 5 min, and immediately centrifuged for 10 min at 11,000×g. The supernatant was decanted, and the cell pellet was stored in the −80 °C. Cell pellets were thawed and incubated with Readylyse Lysozyme, SuperaseIn, Protease K, and 20% SDS for 20 min at 37 °C. Total RNA was isolated and purified using the Qiagen RNeasy Mini Kit columns and following vendor procedures. An on-column DNase-treatment was performed for 30 min at room temperature. RNA was quantified using a Nano drop and quality assessed by running an RNA-nano chip on a bioanalyzer. The rRNA was removed using Illumina Ribo-Zero rRNA removal kit for Gram-negative bacteria. A KAPA stranded RNA-Seq Kit (Kapa Biosystems KK8401) was used following the manufacturer's protocol to create sequencing libraries with an average insert length of around ~300 bp. Libraries were ran on a HiSeq4000 (Illumina). All RNA-seq experiments were performed in biological duplicates from distinct samples.

**ChIP-exo preparation.** To activate each TF, cells were grown on relevant media: M9 minimal medium with 2 g/L glucose and 5 mM methionine for MetJ, M9 minimal medium with 2 g/L glucose and 0.25 mM taurine for CysB, and M9 minimal medium with 3.3 g/L pyruvate for YheO. To identify binding maps for each TF, DNA bound to each TF from formaldehyde cross-linked *E. coli* cells were isolated by chromatin immunoprecipitation (ChIP) with 15 μL specific antibodies that specifically recognize myc tag (9E10, Santa Cruz Biotechnology, Catalog #sc-40), and Dynabeads Pan Mouse IgG magnetic beads (Invitrogen) followed by

stringent washings. ChIP materials (chromatin-beads) were used to perform on-bead enzymatic reactions of the ChIP-exo method[37,62,63]. The sheared DNA of chromatin-beads was repaired by the NEBNext End Repair Module (New England Biolabs) followed by the addition of a single dA overhang and ligation of the first adaptor (5′-phosphorylated) using dA-Tailing Module (New England Biolabs) and NEBNext Quick Ligation Module (New England Biolabs), respectively. Nick repair was performed by using PreCR Repair Mix (New England Biolabs). Lambda exonuclease-treated and RecJf exonuclease-treated chromatin was eluted from the beads and overnight incubation at 65 °C reversed the protein–DNA cross-link. RNAs- and Proteins-removed DNA samples were used to perform primer extension and second adaptor ligation with following modifications. The DNA samples incubated for primer extension as described previously were treated with dA-Tailing Module (New England Biolabs) and NEBNext Quick Ligation Module (New England Biolabs) for second adaptor ligation. Primers are listed in Supplementary Table 8. The DNA sample purified by GeneRead Size Selection Kit (Qiagen) was enriched by polymerase chain reaction (PCR) using Phusion High-Fidelity DNA Polymerase (New England Biolabs). The amplified DNA samples were purified again by GeneRead Size Selection Kit (Qiagen) and quantified using Qubit dsDNA HS Assay Kit (Life Technologies). Quality of the DNA sample was checked by running Agilent High Sensitivity DNA Kit using Agilent 2100 Bioanalyzer (Agilent) before sequenced using HiSeq 2500 (Illumina) following the manufacturer's instructions. ChIP-exo experiments were performed in biological duplicates from distinct samples.

**ChIP-exo processing.** Sequence reads obtained from ChIP-exo experiments were mapped onto the *E. coli* reference genome (NC_000913.3) using bowtie (v1.1.2)[64] with default options in order to generate SAM output files. MACE program[65] was used to define peak candidates from biological duplicates for each experimental condition with sequence depth normalization. Then, each peak was assigned to the nearest operon on either side, using operon definitions from RegulonDB. Only operons 500 base pairs downstream of peak were considered. Final operons on forward strand were required to be in front of the peak, and operons on reverse strand were required to be behind the peak. Genome-scale data were visualized using MetaScope to manually curate peaks (https://sites.google.com/view/systemskimlab/software?authuser=0).

**Compilation of PRECISE.** Raw-sequencing reads were collected from GEO (see Supplementary Data 1 for accession numbers) or produced using the above protocol, and mapped to the reference genome (NC_000913.3) using bowtie (v1.1.2)[64] with the following options "-X 1000 -n 2 -3 3". Transcript abundance was quantified using *summarizeOverlaps* from the R *GenomicAlignments* package (v1.18.0), with the following options "mode = "IntersectionStrict", singleEnd = FALSE, ignore.strand = FALSE, preprocess.reads = invertStrand"[66]. To ensure the quality of the compendium, genes shorter than 100 nucleotides and genes with under 10 fragments per million-mapped reads across all samples were removed before further analysis. Transcripts per million (TPM) were calculated by *DESeq2* (v1.22.1)[67]. The final expression compendium was log-transformed $\log_2(TPM + 1)$ before analysis, referred to as log-TPM. Biological replicates with $R^2 < 0.9$ between log-TPM were removed to reduce technical noise.

**Compilation of the reported *E. coli* regulatory network.** We compiled the global TRN using all interactions from RegulonDB 10.0[3] for both transcription factor and sRNA-binding sites. Binding sites were added from recent ChIP-exo studies[31], in addition to binding sites for Nac and NtrC[68] and potential-binding sites for 10 uncharacterized transcription factors[42]. We also included sigma factor-binding sites, riboswitch information, and transcriptional attenuation from Ecocyc[69]. When reported, mode of effect (i.e. activation or repression) was included. If the effect was unreported, or multiple effects were reported, effects were designated as unknown. All genes absent from PRECISE were removed from the final TRN.

**Computing robust independent components.** We first centered the compendium using wild-type *E. coli* MG1655 grown on glucose M9 minimal media as the reference condition (labeled *control__wt_glc__1* and *control__wt_glc__2*). We subtracted the mean expression of each gene in these two samples from the compendium to calculate log2-fold-change (LFC) deviations from the reference.

We used the Scikit-learn[70] (v0.19.2) implementation of the FastICA algorithm[71] to identify independent components. We executed FastICA 256 times with random seeds, a convergence tolerance of $10^{-8}$, $\log(\cosh(x))$ as the contrast function, and the parallel search algorithm. We set the number of components in each iteration to the number of components that reconstruct 99% of the variance as calculated by principal component analysis (200 components).

The resulting source components (**S**) from each run were clustered using the Scikit-learn implementation of the DBSCAN algorithm[72], with epsilon of 0.1, and minimum cluster seed size of 128 samples (50% of the number of random restarts). DBSCAN does not require predetermination of the number of clusters, and does not require that all points belong to a cluster. The dimensionality of the dataset is therefore estimated by the number of clusters calculated by DBSCAN. The components computed by FastICA are standardized by default, with a mean of 0 and an L2-norm of 1. However, identical components from separate runs may have

opposite signs. Therefore, we used the following distance metric:

$$d_{x,y} = 1 - \left| \rho_{x,y} \right| \tag{1}$$

where $\rho_{x,y}$ is the Pearson correlation between components $x$ and $y$. Each component in a cluster was then inverted if necessary to ensure that the gene with the maximum absolute value in the component had a positive weight, creating sign-consistent clusters. The final independent components were defined as the centroid of each cluster in **S**, and the weightings were defined as the centroid of their corresponding weighting vectors in **A**.

In order to ensure that the final components were consistent across multiple runs, we computed the clustered components 100 times, and found that 92 components were identified in every run ($d_{x,y} < 0.1$ between components), which were the final robust components used in the analysis.

In order to confirm that the i-modulon structure was generally invariant to the composition of the expression compendium, we applied ICA to three subsets of PRECISE. The first subset consisted of data published before January 2018 from unevolved *E. coli* (124 profiles), the second subset consisted of all data published before January 2018 (168 profiles), and the third subset consisted of all data published before January 2018, plus data created to test the i-modulon structure (198 profiles). We compared the resulting components using the absolute value of the pearson correlation coefficient. The resulting network was graphed using the Graphviz[73] python library (v0.9) (Fig. 4b). Correlations below 0.5 were discarded as insignificant.

**Determination of the gene coefficient threshold.** Each component in **S** contains the contributions of each gene to the statistically independent source of variation. Most of these values are near zero for a given component. In order to identify the most significant genes in each component, we modified the method proposed in Frigyesi et al.[74]. For each component, we iteratively removed genes with the largest absolute value and computed the D'Agostino $K^2$ test statistic for the resulting distribution. The D'Agostino $K^2$ statistic is a measure of the skew and kurtosis of a sample distribution[75]. Once the test statistic dropped below a cutoff, we designated the removed genes as significant.

To identify this cutoff, we performed a sensitivity analysis on the concordance between significant genes in each component and known regulons. First, we isolated the 20 genes from each component with the highest absolute gene coefficients. We then compared each gene set against all known regulons using the two-sided Fisher's exact test (FDR < $10^{-5}$). For each component with at least one significant enrichment, we selected the regulator with the lowest *p*-value.

Next, we varied the D'Agostino $K^2$ test statistic from 200 through 1000 in increments of 50, and computed the F1-score (harmonic average between precision and recall) between each component and its linked regulator. The maximum value of the average F1-score across the components with linked regulators occurred at a test statistic of cutoff of 550 (see Supplementary Fig. 5a–c).

Since each set of significant genes represents a set of independently modulated genes, we henceforth refer to these gene sets as i-modulons. Since independent components have no canonical direction, we inverted i-modulons (and related activities) such that the number of positive genes in an i-modulon was always larger than the number of negative genes.

**Associating regulators to i-modulons.** We compared the set of significant genes in each i-modulon to each regulon (defined as the set of genes regulated by any given regulator) using the two-sided Fisher's exact test (FDR < $10^{-5}$). Additionally, combined regulon enrichments were calculated to identify joint regulation of i-modulons (such as NtrC + RpoN and NagC/TyrR), using both intersection (+) and union (/) of up to three regulons. Final i-modulon-regulator associations were determined through manual curation of enriched regulators. Automated characterization of i-modulons for Supplementary Data 2 is described in the Supplementary Methods.

**Cumulative explained variance for ICA.** Components were initially ordered by the L2-norm (sum of squares) of for each row in the **A** matrix for ICA. Cumulative explained variance was calculated for component $K$, as described in the EEGLAB suite:[76]

$$\mathrm{CEV}(K) = 1 - \frac{\mathrm{TSS}\left(\mathbf{X} - \sum_{k=0}^{K} \mathbf{s}_k \mathbf{a}_k\right)}{\mathrm{TSS}(\mathbf{X})}, \tag{2}$$

where TSS(**Y**) is the total sum of squares

$$\mathrm{TSS}(\mathbf{Y}) = \sum_{i,j} \left( y_{i,j}^2 \right), \tag{3}$$

**X** is the original expression profile, $\mathbf{s}_k$ is column $k$ in the **S** matrix, and $\mathbf{a}_k$ is row $k$ in the **A** matrix.

**Comparison of microarray data and PRECISE.** We acquired the microarray compendium from the DREAM5 network inference challenge[10]. We removed expression profiles without biological replicates, and removed expression profiles with an $R^2$ score below 0.9 with its biological replicates. The final microarray

compendium contained 4289 genes and 461 expression profiles. We then randomly selected ten subsets of this compendium, each containing 154 unique conditions to mirror the composition of the PRECISE compendium. We included all experimental replicates of these conditions, resulting in 10 datasets ranging between 255 and 289 total expression profiles. ICA was performed as described above for all 10 datasets.

The microarray dataset from a single research group was acquired from NCBI GEO Series GSE6836[26]. This dataset had similar size to PRECISE (266 experiments) to ensure comparability. Microarray data was processed using the RMA R package[77] (v1.50). ICA was performed as described above for both datasets. PCA determined that 148 components reconstructed 99% of the variance in the microarray dataset, and 104 robust independent components were identified.

Sparse-PCA was performed using the *elasticnet* R package[78] (v1.1.1), searching for the same number of components as with ICA (200 for PRECISE and 148 for microarray data), and a vector of ones as thresholding parameters.

For the comparison figures, each set of components was randomly split into three groups for three-fold validation. For each training set, we selected the top 20 genes in each component and assigned the regulon with the lowest *p*-value from Fisher's exact test (FDR < 1e−5). If no regulon contained significant *p*-values, the component was discarded. We then performed sensitivity analysis for the test statistic cutoff using these regulon assignments, searching for the test statistic value that maximized the F1-score across the training set.

The test statistic trained on the training set was then applied to the testing set to calculate the final validation i-modulons (i.e. significant genes). To assess the similarity between these i-modulons and known regulation, we identified the regulon with the lowest *p*-value (FDR < 1e−5) for each i-modulon in the test set. No manual curation was used to generate the comparison figure.

**Differential activity analysis.** We first computed the distribution of differences in i-modulon activities between biological replicates, and then fit a log-normal distribution to each distribution. We confirmed that the difference in activities between biological replicates followed a log-normal distribution for all i-modulons using the Kolmogorov–Smirnov test and validating through quantile–quantile plots (Supplementary Fig. 5d–f).

To test for differential activity of an i-modulon between two different conditions, we first computed the average activity of the i-modulon between biological replicates, if available. We then computed the absolute value of the difference in i-modulon activities between the two conditions. This difference was compared against the log-normal distribution for the i-modulon to calculate a *p*-value. I-modulons were designated as significant if the absolute value of their activities was >5, and FDR < 0.01.

**I-modulon summation.** We selected samples *control__wt_glc__1* and *control__wt_glc__2* to represent the wild-type cell, and samples *omics__bw_glc__1* and *omics__bw_glc__2* to represent the mutated strains to be corrected. The average activities between replicates were used for the corrections. The corrections were applied to the BW25113 and Thiamine i-modulons.

The i-modulon decomposition is based on the equation

$$\mathbf{X} = \mathbf{SA}, \tag{4}$$

where

$$\mathbf{x}_j = \Sigma \mathbf{s}_i * a_{i,j} \tag{5}$$

for a particular expression profile $j$, where $i$ represents an i-modulon. We aim to produce the correction ($x_2'$) to the expression profile ($x_2$) with respect to a reference expression profile ($x_1$) for all differentially activated i-modulons $i \in$ I:

$$\mathbf{x}_2' = x_2 - \Sigma \widetilde{\mathbf{s}}_i * (a_{i,2} - a_{i,1}), \tag{6}$$

where $\widetilde{\mathbf{s}}_i$ is a vector of zeros except for significant gene coefficients in i-modulon $i$, and $a_{ij}$ is the activity of i-modulon $i$ under condition $j$.

**RNA-seq processing projection for multiple strain comparison.** Raw-sequencing reads and transcriptome abundance were identified similar to as described in the section above, using the following reference genomes: NC_000913.3 (MG1655 and BW25113), NC_007779.1 (W3110), NC_010468.1 (Crooks), NC_012971.2 (BL21(DE3)), NC_009800.1 (HS), NZ_CP008957.1 (O157: H7 EDL933), and NC_004431.1 (CFT073). Genes absent from a particular strain with respect to the reference strains (MG1655) were removed, leaving 915 core genes. We calculated the $\log_2(\mathrm{TPM} + 1)$ values using the same centering to reference conditions (*control__wt_glc__1* and *control__wt_glc__2*) as described above.

Thereafter, we calculated the i-modulon activities for the eight new *E. coli* expression profiles using the previously identified 92 independent components (including all gene coefficients for the 915 conserved genes). We projected the eight new expression profiles (**X′**) onto the previously computed basis (**S**):

$$\mathbf{A}' = pinv(\mathbf{S}) * \mathbf{X}' \tag{7}$$

where **A′** represents the i-modulon activities for the eight strains, and *pinv* is the pseudo-inverse function. This represents the least-squares approximation of **A**.

**Reporting summary**. Further information on research design is available in the Nature Research Reporting Summary linked to this article.

## Code availability

Code central to the conclusions is described in the methods and available at https://github.com/SBRG/precise-db. Additional code is available from the corresponding author upon request.

## Data availability

New RNA-seq and ChIP-exo data reported in this paper are deposited in the NCBI Gene Expression Omnibus with primary accession codes GSE122211, GSE122295, GSE122296, and GSE122320. The complete PRECISE dataset is available in Supplementary Data 1. All other relevant data are available from the corresponding author upon request.

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

## Acknowledgements

We thank Dr. Joe Pogliano for biological insights, Dr. David Heckmann, Dr. James Yurkovich, Dr. Jared Broddrick, Erol Kavvas, Jean-Christophe Lachance, Saugat Poudel, Dr. Adam Feist and Dr. Ke Chen for helpful discussions and Marc Abrams for editorial comments. This research used resources of the National Energy Research Scientific Computing Center, a DOE Office of Science User Facility supported by the Office of Science of the U.S. Department of Energy under Contract No. DE-AC02-05CH11231. This work was funded by the Novo Nordisk Foundation Center for Biosustainability and the Technical University of Denmark (grant number NNF10CC1016517).

## Author contributions

A.V.S. designed the study and performed the analysis, with assistance from L.Y. and B.O.P. Y.G., R.S., Y.H., and S.X. performed expression profiling experiments. Y.G. performed ChIP-exo experiments. D.K. performed expression profiling for multi-strain expression analysis and K.S.C. analyzed the multi-strain data. L.Y. and Z.A.K. provided technical support and conceptual advice. A.V.S. and B.O.P. wrote the manuscript.

## Competing interests

The authors declare no competing interests.
