## [Peer Review File · Nature Communications]

Editorial Note: This manuscript has been previously reviewed at another journal that is not operating a transparent peer review scheme. This document only contains reviewer comments and rebuttal letters for versions considered at *Nature Communications*. Mentions of the other journal have been redacted.

Reviewers' comments:

Reviewer #1 (Remarks to the Author):

This is a resubmission/transfer of a revised manuscript where almost all of my comments have been addressed. As with my first review, I believe that the manuscript is of high quality and useful for our community. A couple issues that need to be worked out before acceptance are the following.

Major:

1. (Lines 64-67) Authors should point to evidence/references supporting this claim: "A major advantage of decomposition-based module detection algorithms, such as singular value decomposition (SVD)²⁰ and ICA, over clustering or network inference methods is that decomposition-based methods simultaneously compute context-specific activity levels for their gene modules."

2. (Lines 161-170): Authors selected COLOMBOS microarray compendium (ref 24) for benchmarking using 10 randomly selected subsets. This is valuable and step towards addressing relevant feedback. However a major issue with COLOMBOS dataset is that gene expression levels are not comparable across multiple experiments (only comparable to the paired reference test experiment). In contrast, PRECISE includes transformation/normalization steps (as described in lines 637-640) under "Computing robust independent components"). This would be affecting the quality of the i-modulons inferred from COLOMBOS dataset. Authors should describe normalizations/transformations steps that are done prior to application of ICA for each dataset to ensure fair comparison. Alternatively use the DREAM5 challenge dataset which is normalized. On a similar note, the manuscript mentions "Biological replicates with $R^2 < 0.9$ between log-TPM were removed to reduce technical noise" (Line 622-624) for the PRECISE dataset. Please clarify in the manuscript whether this step was followed for all datasets. Whenever hyper-parameters (such as $R^2 < 0.9$), authors may perform ICA multiple times with alternative pre-processing steps to select optimal hyper-parameter.

Minor:

1. Figure 1. Caption mentions the term "layer" without clarifying what a "layer" is, please define.

Reviewer #2 (Remarks to the Author):

I reread the paper and the comments to my previous review. I'd like to emphasize the fact that I do think that this is a nice study and that it is clear to me that much work has been spent on it. I also believe that the study, data, and the results will be useful for other researchers. I still do not feel that the paper should be published in a more specialized journal than Nature communication.

Please upload all the code also to the journal as a supplementary.

Reviewer #3 (Remarks to the Author):

(Note that I was Reviewer #3 for the manuscript version that was previously submitted to **[REDACTED]**)

The authors addressed the main concern I had regarding the validation of the i-modulons. I however still believe that the claim that the E. coli transcriptome consists of independently regulated modules (which is actually the title of the paper) is too strong. Almost 1/3 of the variance remains unexplained and the authors themselves found out that some i-modulons are regulated by multiple regulators. While I think that the manuscript and work within are of high merit, at least the title should be slightly modified (maybe in something like "The E. coli transcriptome *mostly* consists of independently regulated modules).

Typos:

- Line 142: two (instead of four) were linked to transcriptional attenuation...
- In line 677, the authors refer to Figure S3c, which is not the correct figure. Actually, I could not find the figure that compares the components obtained from the three subsets of PRECISE.

Response to Reviewers:

Changes to the manuscript are marked in red if the change was directly due to a reviewer comment. Additional changes are marked in blue for minor changes in wording to fix grammar or comply with the Nature Communications manuscript checklist.

Reviewer #1 (Remarks to the Author):

This is a resubmission/transfer of a revised manuscript where almost all of my comments have been addressed. As with my first review, I believe that the manuscript is of high quality and useful for our community. A couple issues that need to be worked out before acceptance are the following.

Major:

1. (Lines 64-67) Authors should point to evidence/references supporting this claim: “A major advantage of decomposition-based module detection algorithms, such as singular value decomposition (SVD)²⁰ and ICA, over clustering or network inference methods is that decomposition-based methods detect gene modules while simultaneously computing the context-specific activity levels for these gene modules.”

In order to clarify this claim, we moved the Alter et al 2000 citation to the end of this statement, and included a citation to the Saelens et al review, which provides detailed explanations of 42 module detection methods in the supplement (Lines 67-71). Both of these papers state that matrix decompositions reduce the expression matrix into two matrices, one relating to genes (Alter et al calls these “eigengenes”), and one relating to samples (Alter et al calls these “eigenarrays”). In contrast, clustering and network inference methods only provide relationships between genes, and cannot intrinsically provide activity levels for transcription factors without the use of external tools.

None of the 37 clustering and network inference methods described in the Saelens et al review describe the ability to compute context-specific activities. However, this is explicitly described for matrix decompositions in the supplement of that review (page 42).

2. (Lines 161-170): Authors selected COLOMBOS microarray compendium (ref 24) for benchmarking using 10 randomly selected subsets. This is valuable and step towards addressing relevant feedback. However a major issue with COLOMBOS dataset is that gene expression levels are not comparable across multiple experiments (only comparable to the paired reference test experiment). In contrast, PRECISE includes transformation/normalization steps (as described in

lines 637-640) under “Computing robust independent components”). This would be affecting the quality of the i-modulons inferred from COLOMBOS dataset. Authors should describe normalizations/transformations steps that are done prior to application of ICA for each dataset to ensure fair comparison. Alternatively use the DREAM5 challenge dataset which is normalized. On a similar note, the manuscript mentions “Biological replicates with $R^2 < 0.9$ between log-TPM were removed to reduce technical noise” (Line 622-624) for the PRECISE dataset. Please clarify in the manuscript whether this step was followed for all datasets. Whenever hyper-parameters (such as $R^2 < 0.9$), authors may perform ICA multiple times with alternative pre-processing steps to select optimal hyper-parameter.

Thank you for your suggestion. We have performed ICA on the DREAM5 compendium and updated Figure 2 accordingly (replacing the COLOMBOS bars). As described in the methods (Lines 613-617), we removed experiments from the DREAM5 compendium without replicates, and only kept replicates with an $R^2 > 0.9$ between replicates. The results still show that ICA of the RNA-seq compendium outperforms ICA of all microarray datasets.

Minor:

1. Figure 1. Caption mentions the term “layer” without clarifying what a “layer” is, please define.

Thank you for mentioning this. We have updated the caption (line 934-937) to read:

“These coefficients are scaled by the component’s condition-specific activities (row in A) to form the component’s contribution to the transcriptomic compendium (e) The sum of the contributions from the 92 components reconstructs most of the variance in the original compendium.”

Reviewer #2 (Remarks to the Author):

I reread the paper and the comments to my previous review. I'd like to emphasize the fact that I do think that this is a nice study and that it is clear to me that much work has been spent on it. I also believe that the study, data, and the results will be useful for other researchers. I still do not feel that the paper should be published in a more specialized journal than Nature communication.

Please upload all the code also to the journal as a supplementary.

Thank you for these comments. We have included additional code in the github repository: <https://github.com/SBRG/precise-db>, which is references in the Code Availability section and on line 160.

This code runs ICA from an expression compendium and calculates the i-modulon thresholds.

Reviewer #3 (Remarks to the Author):

(Note that I was Reviewer #3 for the manuscript version that was previously submitted to **[REDACTED]**)

The authors addressed the main concern I had regarding the validation of the i-modulons.

I however still believe that the claim that the E. coli transcriptome consists of independently regulated modules (which is actually the title of the paper) is too strong. Almost 1/3 of the variance remains unexplained and the authors themselves found out that some i-modulons are regulated by multiple regulators. While I think that the manuscript and work within are of high merit, at least the title should be slightly modified (maybe in something like “The E. coli transcriptome *mostly* consists of independently regulated modules).

As per your suggestion, we have modified the title to read: “Machine Learning Decomposes the Escherichia coli Transcriptome into Independently Modulated Groups of Genes”

Typos:

- Line 142: two (instead of four) were linked to transcriptional attenuation...

We have changed the wording to clear up some ambiguity for that section, and fixed a numerical error in the pie chart itself (Number of single TFs should be 40).

- In line 677, the authors refer to Figure S3c, which is not the correct figure.

Actually, I could not find the figure that compares the components obtained from the three subsets of PRECISE.

The correct figure is Figure 4b. This has been updated in the manuscript.

REVIEWERS' COMMENTS:

Reviewer #1 (Remarks to the Author):

All points have been addressed by the authors.